# Integrating Ecosystem Vulnerability in the Environmental Regulation Plan of Izmir (Turkey)—What Are the Limits and Potentialities?

Stefano Salata *, Sıla Özkavaf-Şenalp and Koray Velibeyoğlu

Laboratory of Ecosystem Planning and Circular Adaptation—Lab EPiCA, Department of City and Regional Planning, Izmir Institute of Technology, Gülbahçe Kampüsü Urla, İzmir 35430, Turkey; silaozkavaf@iyte.edu.tr (S.Ö.-Ş.); korayvelibeyoglu@iyte.edu.tr (K.V.)
* Correspondence: stefanosalata@iyte.edu.tr

**Abstract:** The land-use regulatory framework in Turkey is composed of several hierarchical plans. The Environmental Regulation Plan pursues comprehensive planning management, which ranges between 1/100,000 and 1/25,000 and defines the framework for local master plans. Unfortunately, there is scarce knowledge of how these plans effectively protect the environment. Besides, these plans have poor consideration of socio-economic dynamics and the ecosystem vulnerability, while evaluating the actual conflicts or synergies within the localization of ecological conservation and settlement expansion areas. In this work, an ecosystem-based geodatabase was created for the western Izmir area (Turkey). The dataset has been created by employing a supervised classification sampling of Sentinel-2 images acquired on 28 March 2021, while accessing ONDA-DIAS services to L2C products. Then, the InVEST software was used to map the Habitat Quality and the Habitat Decay, while the ArcMap raster analysis tool was employed to generate the Normalized Difference Vegetation Index. The results were used to classify the ecosystem vulnerability of the western metropolitan area of Izmir and then superimposed to the Environmental Regulation Plan of the city of Izmir (2021), thus evaluating synergies and conflicts. Although integration of the ecosystem services approach into spatial planning is lacking in the planning practice of Turkey, the paper provides an operative methodology to integrate ecosystem evaluation in environmental planning as a basic strategy to support sustainable development.

**Keywords:** environmental planning; ecosystem services; GIS; vulnerability; biodiversity; landscape protection

## 1. Introduction

### 1.1. A General Overview of Izmir Metropolitan City

Although the Izmir Peninsula has a special landscape that has been blended by human–environment interaction over the centuries [1] composed of natural and semi-natural Mediterranean vegetation covering the promontory on the Aegean Sea [2], the metropolitan city of Izmir has been undergoing a massive urbanization process that threatens the natural environment [3].

As a result, the region is experiencing one of the most rapid dynamics of transformation [4]: the urbanization and suburbanization process is occurring with low planning control and difficult coordination between local development plans and general urban planning guidelines [5].

To what concerns the planning system in Turkey, environmental regulation plans are general plans prepared on the scales of 100,000 and 25,000. However, the preparation of upper-scale plans for Izmir has been problematic since the 1980s. Plans had been subjected to multiple amendments because of changes in regulatory frameworks. Inconsistency in

the upper-scale planning process has resulted in disharmonized lower-scale planning and implementation.

The rapid transformation and lack of coordination are often accompanied by insufficient knowledge of the notions of ecosystem vulnerability and the value of habitat quality [6,7]. Despite some pioneering experiences [8], the integration of ecosystem services approaches in the planning processes is lacking in the Turkish planning experience. Therefore, the need to integrate the decision-making process with an environmentally sound perspective is highly demanded [9,10].

First, it is crucial to describe the framework of how planning institutions operate in Turkey. Different authorities are entitled to prepare strategic and spatial plans depending on scale and coverage. The Presidential Office coordinates the national Development Plan, the Ministry of Environment and Urbanism prepares the National Spatial Strategy Plan and Environmental Regulation Plans on provincial and regional scales ranging from 1/100,000 to 1/25,000. Although not an administrative tier, Regional Development Agencies prepare regional economic plans on the NUTS 2 level. Metropolitan Municipalities prepare master development plans and district municipalities prepare implementation plans.

In Turkey, there is a two-tiered municipal structure in a metropolitan area: metropolitan municipalities and district municipalities. Metropolitan areas are governed by law no. 6360 enacted in 2012. According to the law, the responsibility of these metropolitan municipalities (30 all over in Turkey) covers provincial borders. In terms of the planning authority, metropolitan municipal councils prepare and approve upper-scale spatial plans such as master plans on the provincial level; district municipalities are entitled to produce lower scale plans such as implementation plans. Law no. 6360 was built upon two previous laws on metropolitan areas: Law no. 3030 (1984) and Law no. 5216 (2004). In each law, metropolitan area borders were expanded to provincial borders. In this process, new surrounding districts were included in the metropolitan area of Izmir. For example, while Urla, Güzelbahçe and Seferihisar districts are included in metropolitan borders with the law no. 5216; the Karaburun and Çeşme districts were included in 2012 (the inclusion process ended after 2014). In addition, the most important decision of the law was changing the status of rural villages into urban neighborhoods within metropolitan area borders. This situation has been criticized for creating blurred boundaries between rural and urban areas, enabling expansion of urban functions in rural areas and a drastic increase in urban population [11].

By 2011, the organization and competencies of the Ministry of Environment and Urbanism were reorganized to merge the responsibilities of several authorities to coordinate urbanization and environmental issues. This reorganization can be considered as the centralization of authority on the one hand and the unification of planning processes on the other.

It is also important to mention that administrative institutions have been undergoing a transformation that led to the merging and closure of authorities such as ministries, municipalities and directorates. For example, while the General Directorate for Protection of Natural Assets works under the Ministry of Environment and Urbanism, previously it had been operating under the Ministry of Culture and Tourism since 2011. This directorate governs Special Environment Protection Areas (SEPA), Natural Protection Sites (1st, 2nd and 3rd degree) and Natural Assets (caves and monumental trees). This means various types of natural areas that are regulated by many laws and different authorities. Besides, the General Directorate of Nature Conservation and National Parks under the Ministry of Agriculture and Forestry is responsible for areas with the status of National Parks, Natural Parks, Natural Monuments, Nature Protection Areas, Wetlands and Wildlife Improvement Areas. In 2019, natural protection sites received different statuses such as "Sensitive Area in Primary Protection", "Qualified Natural Protection Areas" and "Sustainable Conservation and Controlled Use Area".

Overall, the planning institution in Turkey has been balancing the demands of market institutions and necessities of public interest in planning. Without having a comprehensive

upper-scale plan that aims to guide local development and define natural protection, urban development has been coordinated by a piecemeal planning approach that leads to an unintegrated and uncontrolled built environment.

### 1.2. Literature Review

The approach we used can be collocated among the experimental works which want to define new tools and methods for supporting the decision making of environmental planning at different scales [12–14]. Particularly, we intend to bridge the gap between theory and the practical utilization of the Ecosystem Service in real planning documents, as indicated by Costanza et al. (2017), where the revision of the ES cascade model shows how spatial modelling constitutes a basic pre-condition for achieving the sustainability of plans and projects at different scales [15,16]. However, the application of the ES vulnerability concept to ecosystems is still an emerging topic.

As stated by Weißhuhn et al. (2018), proper ecosystem management should be based on an ecosystem's vulnerability analysis, which includes the capacity to recover rapidly from man-made and natural disturbances [17,18]. According to the United Nations Intergovernmental Panel on Climate Change (2014), vulnerability is the predisposition of a system to be damaged by an external event [19]. Its main components are divided into sensitivity (characteristics of the system) [20] and coping capacity (capacity to resist and absorb shocks) [21].

One of the first holistic studies on environmental vulnerability was developed in 2004 by the South Pacific Applied Geoscience Commission in cooperation with the United Nations Environment Program. More than 50 indicators were calculated globally [22]. Still, the concept of ecosystem vulnerability gained visibility only after 2010, when the implementation of the Ecosystem Service approach has been widely diffused among researchers [23–25]. More recently, the ecosystem vulnerability has been re-defined in the contexts of the Anthropocene [26], where complex systems of anthropic, natural and seminatural areas interact simultaneously with different degrees of dominance [27–29]. Within these paradigms, the studies on ecosystem vulnerability gained importance in the context of the resilience of Socio-Ecological and Technological Systems (SETS) [30]. According to this vision, the first step toward a resilient system is the reduction of its intrinsic vulnerability to any kind of hazard [31–34].

An ecosystem vulnerability assessment could be used to estimate the inability of an ecosystem to tolerate stressors over time and space [35]. Maps of vulnerable areas can be employed to define proper management/conservation zones in different SETS [36].

As for its past implementation, ecosystem vulnerability analysis has been mostly applied in conservation biology [37], climate change [38] and ecological risk assessments [39], while a limited diffusion has been reached on landscape planning. However, nowadays, ecosystem resilience is widely diffused on Geographic İnformation System applications that apply multicriteria analyses to map the quality of ecosystems and their predisposition to be damaged by multiple hazards [40,41].

Here, we want to define a methodology to inform planning processes through ecosystem vulnerability data. In particular, we intend to support and integrate Environmental Regulation Plans considering the ecosystem vulnerability distribution [6,35] and evaluate the real conflicts or synergies within the localization of ecological conservation areas [5,42,43].

### 1.3. Aims of This Study

Ecosystem mapping assessment in Tukey is not yet currently practiced due partly to the paucity of production and distribution of digital data for soil ecosystem mapping and partly due to a weak cultural approach to integrating ecological values in planning [44].

Therefore, in this research, we applied a pioneer experimental methodology of Ecosystem Service integration with the documental maps of the existing planning regulation system in the western metropolitan area. We developed an ad hoc methodology that employed the most advanced Ecosystem Service modelling by Integrated Valuation of

Ecosystem Services and Tradeoffs—InVEST (Habitat Quality and Decay) [45–47], and we integrated the results with the Normalized Difference Vegetation Index (NDVI) produced for the same catchment [48–50]. GIS modelling and ecosystem mapping rely on an auto-produced Land Use Land Cover (LULC) [51,52] of the western Izmir area employing a supervised classification sampling method to a Sentinel Copernicus image using ArcGIS (ver.10.8.1) [53]. We created, for the first time, a new Ecosystem Vulnerability index for the western Izmir area to support the existing planning framework and evaluate its primary purposes to define the conservation zone and indicate the new expansion areas for settlements.

The paper is structured as follows: Section 2 (methodology) explain the characteristics of the Area of Interest (AOI) and the GIS processes employed to map the HQ, the Decay, the NDVI and the conservation/expansion zones of the Environmental Order Plan of İzmir. Section 3 briefly presents the results of the Ecosystem Vulnerability analysis, while in Section 4, the results are discussed against validation of the conflicts and synergies between the ecosystem vulnerability and the Environmental Regulation Plan. Section 5 summarizes the conclusions of this work, emphasizing the main innovations.

## 2. Materials and Methods

### 2.1. The Physical, Social and Economic Dynamics in the Area of Interest (AoI)

Raw data on the recent land-use change analysis between 1990 and 2018 [54–56] based on the Corine Land Cover dataset [57] in the province of Izmir (hereafter Izmir metropolitan city) demonstrate that more than 33 thousand hectares of land were converted from agricultural or natural/semi-natural into urban uses at a speed of urbanization equal to 4.26 square meters for each resident per year, while provoking a sensible reduction of the ecological integrity of this part of the Aegean Promontory (+99% urban areas). The urbanization process occurred at the expense of plain and fertile agricultural areas (26 thousand hectares). However, the same process occurred even at the expense of the characteristic natural and semi-natural Mediterranean environment surrounding this part of Turkey [44,58]: more than ten thousand hectares of semi-natural land uses disappeared in the last 28 years while determining a strong biodiversity reduction process. As several authors demonstrate [59–61], Corine Land Cover data underestimate the real process of urbanization for many technical reasons [62]. Therefore, the above-mentioned numbers outline that the metropolitan city of Izmir has been subjected to rapid urbanization that threatens the natural environment coping with an increase in population by 60.3% since the last 20 years [63].

The Area of Interest (AoI) of this study forms a part of the extended administrative boundary of the Izmir metropolitan city. In particular, the selection comprises the southwestern districts of the metropolitan city, spanning 194,699 ha and including the following districts: Balçova, Çeşme, Güzelbahçe, Karaburun, Narlidere, Seferihisar and Urla. These districts range in nature from urban and peri-urban to the most touristic and naturalistically attractive zones of the metropolitan area (i.e., Cesme and Karaburun). The extension of the AoI was included in the tile size of Sentinel L2A downloaded from the Copernicus ONDA DIAS platform.

The AoI is characterized by a heterogeneous morphological condition, with an extended peri-urban system that is developed along the western coast. The mountains lie perpendicular to the coastline, creating a scattered and discontinuous settlement pattern. Lowlands and coastal lines leave space for settlement, yet rapid urbanization leads to settlement expansion on mountain slopes and productive lands. The area shows a typical subhumid Mediterranean climate, and its vegetation is formed mainly by forests and scrubs accompanied by groves and rangelands.

According to our study (see next chapter), the LULC composition shows that the most abundant part of the area is covered by natural or semi-natural land (40% low vegetation land and 37% natural areas), while the agricultural areas occupy 10% and paved roads

and other impervious areas occupy 8%. Industry occupies less than 1% and water bodies occupy 2.5% of the AoI (Figure 1).

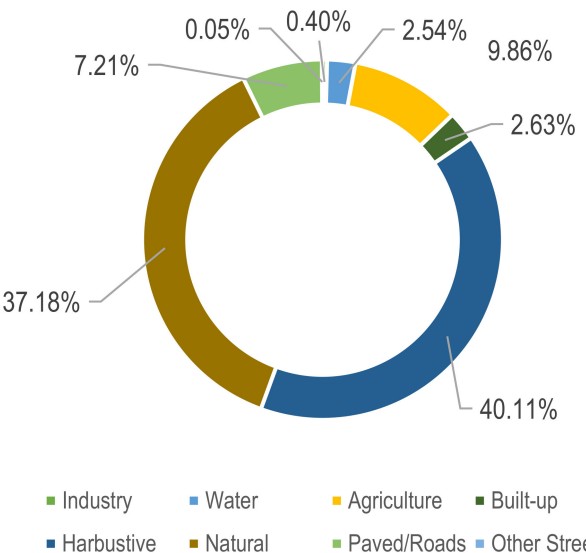

**Figure 1.** Land-use Land Cover Composition in the Area of Interest.

Overall, anthropic-related uses occupy more than 10% of the AOI while urgently requiring the need to govern and steer the incoming urban development process (see Figure 2).

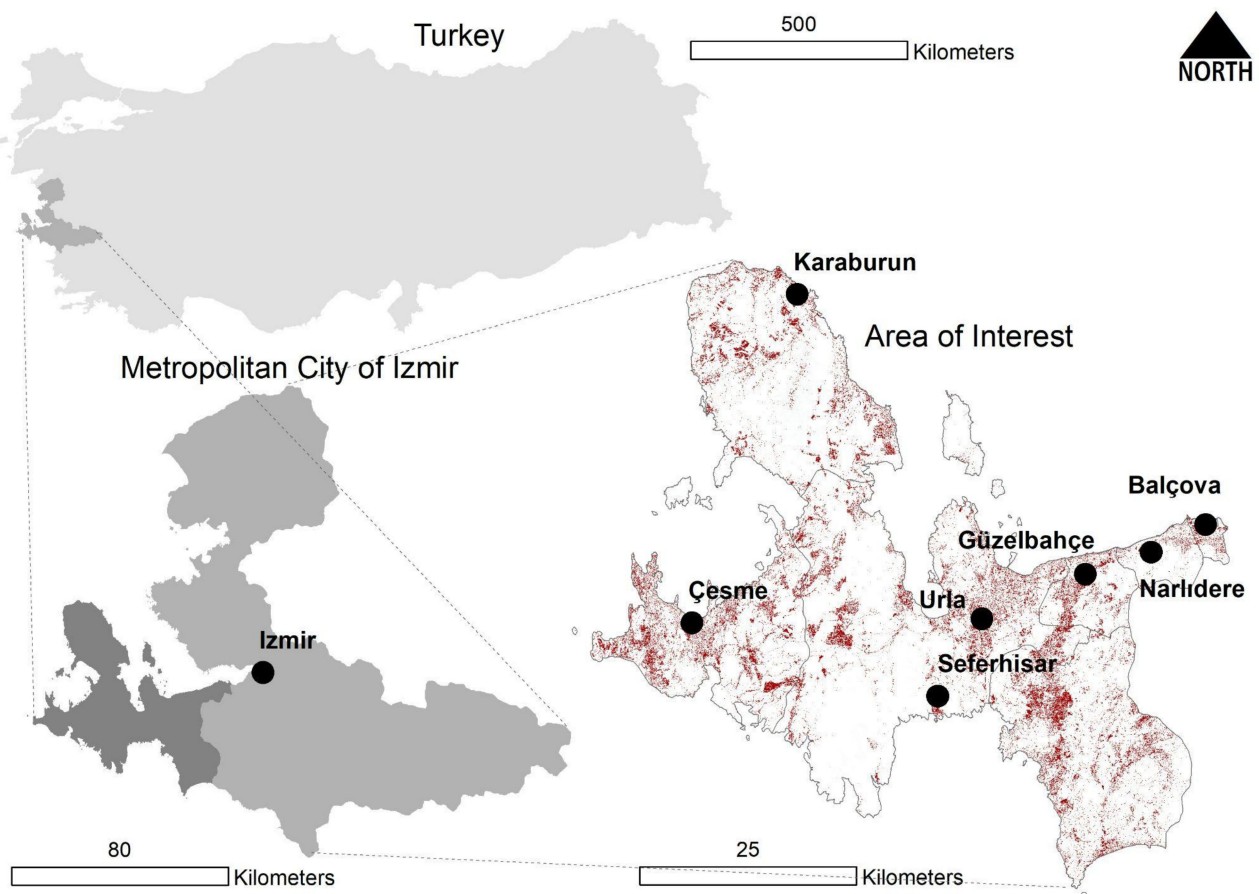

**Figure 2.** Localization of the Area of Interest.

Currently, the overall population of the AoI is 356.001, and it has increased by 63% in the last 20 years [63]. Urbanization processes in the Peninsula have been triggered after the construction of the Izmir–Cesme motorway during the first half of the 1990s. The highway starts from Balçova and connects each district of the Peninsula with the city. It cuts the AoI horizontally from east to west and negatively influences the ecological integrity of the landscape [2].

With increased accessibility, the population distribution in the metropolitan city has drastically changed with the migration from the center to the peripheral districts. While central districts such as Balçova and Narlıdere represent lower rates of population increase, they hold 40% of the total population of AoI due to the earlier inclusion of these districts in the metropolitan city (see Figure 3). Improved mobility enabled commuting from Güzelbahçe and Urla districts, where the settlement areas continuously extend from the center of Izmir. They became suitable locations for residents who demand a suburban life, more connection with nature and sea and access to the new housing stock. Their peculiarity is represented by an exceptional growth rate and the growing role of polarity in the metropolitan urban network supported by planning decisions [64].

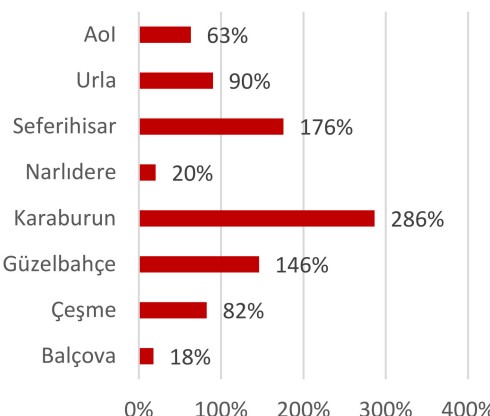

**Figure 3.** Population changes in the districts of AoI (2000–2020). Source: TurkStat, 2021.

Being a popular summer location for Izmir residents is reflected in secondary housing that surrounds especially the shores of the Urla, Karaburun, Çesme and Seferhisar districts. Therefore, the summer population in the Peninsula differs from the winter population due to being a seasonal location. Obviously, the demand for new houses resulted in a sharp increase in housing and land prices. According to the Global Residential Cities Index (2020) report that monitors residential prices across 150 world cities, Izmir has been listed as the second city where the housing prices increased the most, with a rate of 27.8% compared with the previous year [65]. This process, coupled with "amenity migration" for newcomers (a trend that can be observed in especially Urla), is represented by online market research demonstrating that the price for the land with residential permits per sqm unit has increased by 162.48% in Urla for the last four years [66].

Cesme is another typical example of being the most important tourist area of Izmir. It has always been a popular tourism destination and home for summer houses, but the opening of the motorway significantly increased the accessibility to the district and triggered tourism development. The population of Cesme is reported to be 46,093 as of 2021, and its population increased by 82% in the last 20 years (TurkStat, 2021). It should also be noted that the summer population of the district can be 40 times greater than the winter population [2]. In 2019, 16,624 ha of area in the district was announced as "Cesme Culture and Tourism Conservation and Development Zone" by a Presidential Decision. The main problem is that the project offers land-use expansions and fast population projection that contradicts upper-scale plans such as 1/100,000 Environmental Regulation Plan and 1/25,000 Master Development Plans.

This polycentric socio-economic and ecological system also inhabits a network of mountain, seaside and lowland villages holding different livelihood opportunities with diverse natural and environmental assets. While Seferihisar, a CittaSlow town, is home to secondary housing in the shoreline and productive landscape composed of olive groves in the mountainous parts and agricultural areas on the lowlands, the Karaburun Peninsula displays an example of a protected mountainous landscape and seascape. The district has distinct geography with rich terrestrial and aquatic biodiversity and natural and archaeological protection areas, some of which are under protection by international agreements such as the Bern Convention and Cities. Topography does not allow agricultural activities; therefore, olive growing, farming and fishery have been the common source of livelihood. In 2019, the Karaburun district and some parts of the Cesme district (an area covering 946.56 km$^2$) were given the status of "Karaburun-Ildır Bay SEPA" by the Ministry of Environment and Urbanism.

### 2.2. Planning Processes and Evaluation of Environmental Regulation Plan

The study area comprehends the most significant proportion of protected areas in İzmir as defined by the Presidential decision (see Figure 4 left). Apart from the efforts of the Ministry, a nonbinding regulatory study of a national NGO, Nature Society, determined the Key Biodiversity Areas (KBA) all around Turkey. The following KBAs fall into the borders of AoI. These are Cesme Western Promontory with 3465 ha, Alacatı in Cesme with 56,759 ha, Karaburun and Ildırı Bay Islands with 87,274 ha, Cicek Islands of Urla with 8718 ha and Doganbey Shores in Seferihisar with 7465 ha.

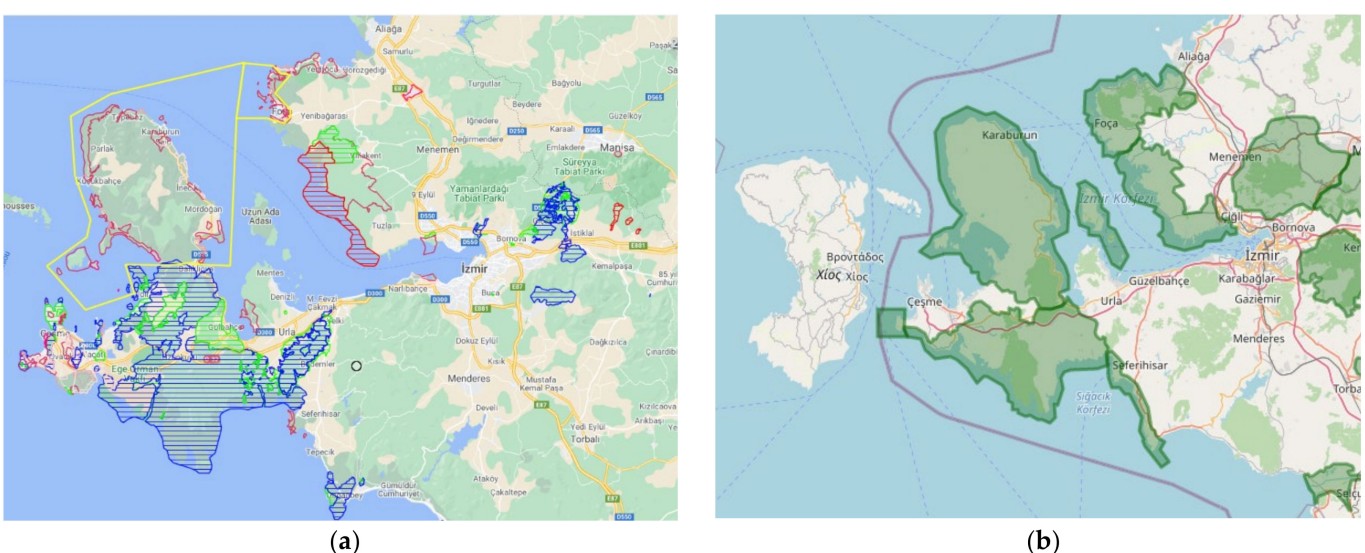

(**a**)                                                                                           (**b**)

**Figure 4.** (**a**) Regulated Protection Areas in Izmir (1), (**b**) Key Biodiversity Areas in the AoI (2). source: (1) https://says.csb.gov.tr/citizen (accessed on 20 August 2021) (2) https://umap.openstreetmap.fr/tr/map/turkiye-onemli-doga-alanlar_555075#9/38.4590/26.8423. (accessed on 9 September 2021).

Between 1973 and 2007, the future of the Izmir metropolitan area was guided through 1/25,000 Master Development Plans without upper-scale comprehensive regional and landscape vision. At the same time, districts prepared 1/5000- and 1/1000-scale implementation plans and their amendments. This situation created unplanned development and an uncoordinated natural conservation process occurring in a non-holistic manner for the Izmir Metropolitan Area.

In 2007, the first 1/100,000 ERP was prepared for the Izmir–Manisa–Kütahya region and then cancelled in 2011 upon objections from municipalities, professional organizations and NGOs. Between 2011 and 2014, no ERP guided lower-scale development plans. A new 1/100,000 ERP was prepared in 2014 with new borders covering Izmir and the Manisa region forwarding 2025. However, the plan was also subjected to objections and criticisms

concerning the planning process and its decisions [67]. Between 2014 and 2020, the plan was subjected to 19 revisions: 3 overall, 7 parcel-based, 1 plan note and 8 areal based. The AOI was subjected to 3 amendments in this context: two overall and one areal revision.

In any case, the 2014 ERP was replying to an almost similar plan approved in 2009 (despite the exclusion of Kütahya). The plan and its revisions were criticized for being prepared using old data, not realizing the objections that have led to the cancellation of the previous plan and for utilizing the landscape scale of parcel-based digital layers that were obsolete [67]. Public participation is another issue, as NGOs and professional organizations are excluded from the planning process. The plan suggests a population increase for the Izmir metropolitan area with 7,424,000 by 2025, which was blamed for being a number beyond natural increase and correlation of expansion areas for the increased population. In the plan, the population projection in Urla is expected to be almost double, while Seferihisar is expected to increase by 120% in 2025. Another problem is the use of ambiguous land-use types such as "Preferred Area of Use", "Special-Crop areas" or "Agritourism" which include complimentary commercial activities and social and technical infrastructure.

Lastly, the identification of protection zones in 1/100,000 ERP does not recognize the real-time state of the environment and does not classify the major pressures (threats) through which the environmental system is affected. On the other hand, a modification in upper-scale plans results in revisions in lower-scale plans that create a vicious cycle and fuel unregulated development that enables interference of market institutions and land speculation.

### 2.3. Production of Land-Use Land Cover and Normalized Difference Vegetation Index

One of the causes of the inefficacy of targeting conservation areas in the ERP was the utilization of digital GIS processing without an updated and real-time digital knowledge of the state of the environment. An Ecosystem Service modelling session has been set up to overcome this limit. Nonetheless, Ecosystem Service modelling is highly dependent on LULC input, and often the utilization of these datasets is limited by the obsoleteness of data. To produce and update LULC for the AoI, the Supervised Classification Sampling of ArcGIS (ver.10.8.1) to original Copernicus images was applied. The characteristics of the acquired image are synthesized in Table 1.

**Table 1.** Characteristics of the originally downloaded Sentinel image.

| Product Name | Creation Date | Size | Instrument | Processing Level | Product Type |
|---|---|---|---|---|---|
| S2B_MSIL2A_20210328T085559_N0214_R007_T35SNC_20210328T113525.SAFE_20200928T131819 | 28 March 2021 | 1.01 GB | MSI Multi-Spectral Instrument | 2A | S2MSI2A |

We accessed the ONDA-DIAS platform within the personalized user's account, then the research query "L2" was inputted in the product list while selecting the most recent images. We chose the most recent cloud-free image from the automatic selection by using the preview function. Then, all the spectral bands were downloaded with a ground resolution of 10 m per pixel.

For this work, only four out of the thirteen original bands were employed in the analysis: the three visible bands plus the Near-Infrared were used to generate the Normalized Difference Vegetation Index (see Table 2).

**Table 2.** Information on Band composition.

| Band | Resolution | Central Wavelength | Description |
|---|---|---|---|
| B2 | 10 m | 490 nm | Blue |
| B3 | 10 m | 560 nm | Green |
| B4 | 10 m | 665 nm | Red |
| B8 | 10 m | 842 nm | Visible and Near Infrared (VNIR) |

The conversion from multiple bands to a single band image was conducted using the raster composite band's tool, which output constituted the baseline layer employed for the LULC-supervised classification.

Then, a supervised classification process was initiated using the training sampling features. Land uses were classified as follow:

- Urban (15 samples);
- Industry/High impermeable urban layers (10 samples);
- Agricultural Land (10 samples);
- Water (15 samples).
- Streets (7 samples);
- Shrubs (10 samples);
- Natural (7 samples);
- Barren/Rock (15 samples).

Results of the supervised classification sampling are reported in Figure 5.

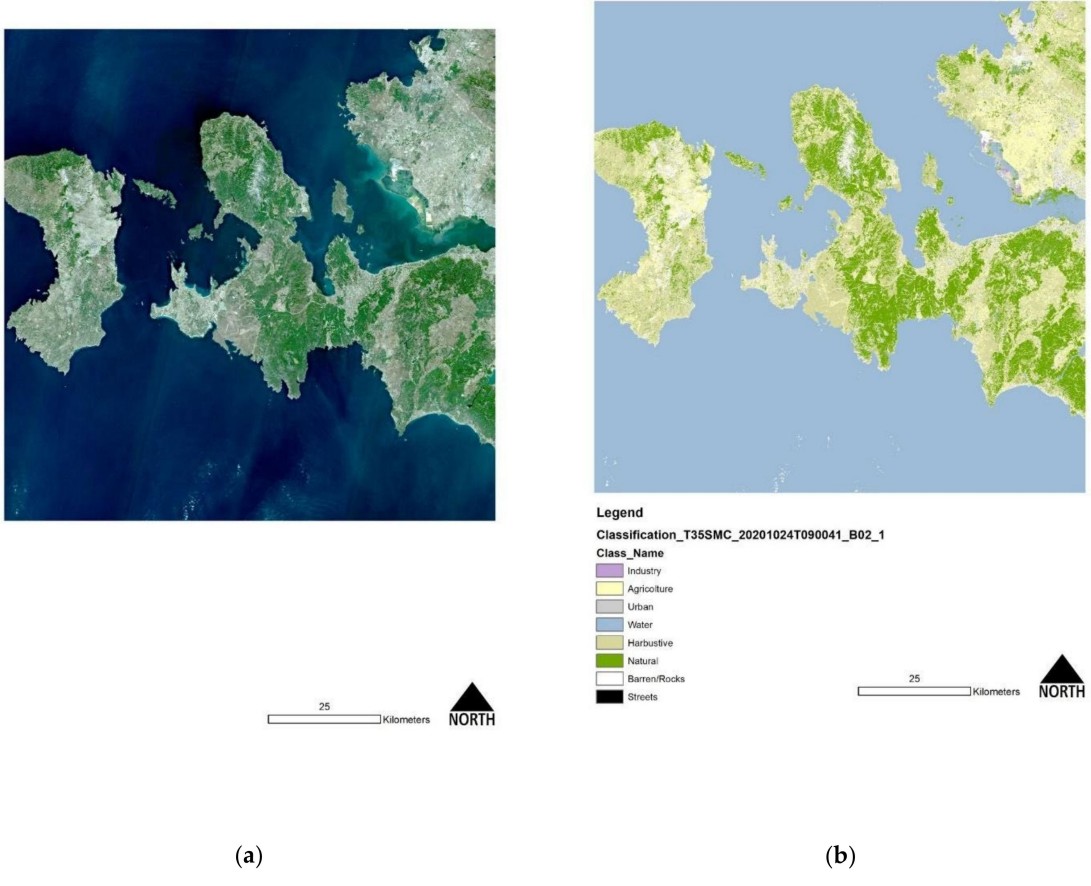

(**a**)　　　　　　　　　　　　　　　　　　　(**b**)

**Figure 5.** (**a**) original RGB composite bands, (**b**) post-processed land-use Land Cover by supervised classification.

After a visual check, we made some post-processing adjustments using the GHSL layers to obtain a final reliable final classification:

1. We combined the Digital Elevation Model and GHSL (the urban density Layer);
2. We delated the urban built-up land above 500 m of altitude;
3. We combined our supervised classification with the urban density layer while removing incoherencies by visual checking;
4. We exported the new file usi5g raster lookup.

The final classification has been finally visually supervised by customer validation using Google Earth and the ESRI Imagery Basemap imported in the GIS project in the

AoI. The process has been carried out manually since the LULC dataset was relatively small. The only official digital LULC classification for this area is Corine Land Cover (2018), whose scale of representation and geometrical precision is far from any other reasonable utilization for this specific study [44,61,62].

As earlier introduced, the image classification has also been used to set the NDVI index. The NDVI measures vegetation quality by detecting the quantity of chlorophyll in plants [48,49]. Healthy green areas strongly absorb visible light, while the leaves reflect near-infrared light.

NDVI has been created by using the "Image Analysis" tool; a new final raster with the NDVI properties has been produced using the Infrared (B8) and Red-visible (B4) Bands (see Table 2) while obtaining the final output (see Figure 6).

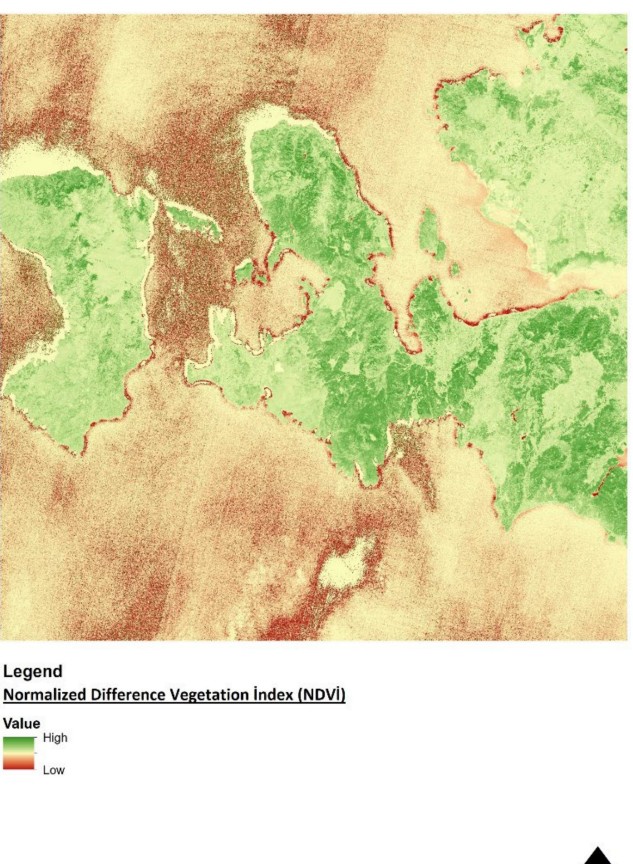

**Figure 6.** The Normalized Difference Vegetation Index.

*2.4. ES Processing*

As previously mentioned, Habitat Quality and Habitat Decay were spatially mapped using the last available release of the software Integrated Evaluation of Ecosystem Service and Tradeoff (InVEST) produced by Stanford University for the Natural Capital Program [68,69].

The model works with relatively few inputs: a LULC dataset (raster), the threats (raster) and two biophysical excel tables inputted in .csv format that contains the interaction between the habitats and their threat: the table of threats which assign a weight to each identified threat and a sensitivity table for each LULC to the selected threats (see Tables 3 and 4).

A unique raster file named "urban" grouped the streets and all urban/industrial zones to simplify the modelling process.

The values of input tables (Tables 3 and 4) were assigned using reference data during the LIFE SAM4CP research.

**Table 3.** Habitat Quality modelling. Threat values in the input .csv table. The abbreviation MAX_DİST stays for the maximum distance at which the threats generates a decay on habitats, while CUR_PATH stays for the file path where the current land-use scenario is saved in the system.

| THREAT | MAX_DIST | WEIGHT | DECAY | CUR_PATH |
|---|---|---|---|---|
| urban | 0.8 | 0.9 | linear | threat/urban.tif |

**Table 4.** Habitat Quality modelling. Sensitivity values for each Land Use in the input .csv table.

| LULC | NAME | HABITAT | Urban |
|---|---|---|---|
|  | Urban | 0.05 | 0 |
| 0 | Industry | 0 | 0 |
| 35 | Streets | 0 | 0 |
| 230 | Barren | 0.40 | 0.6 |
| 100 | Agriculture | 0.5 | 0.7 |
| 177 | Harbustive | 0.8 | 0.9 |
| 211 | Natural | 0.95 | 1 |
| 9 | Water | 1 | 1 |

## 3. Mapping Results

### 3.1. The Ecosystem Vulnerability Map

The spatial indicators introduced in Sections 2.3 and 2.4 are complementary but different as they gather different environmental sides, thus resulting as extremely useful to define the vulnerability of ecosystems. The Habitat Quality provides the essential value of the ecosystem integrity or biodiversity potential. At the same time, the Decay adds important information concerning the threat to which the specific habitat is affected (see Figure 7). Finally, the NDVI can be considered as supporting data that reveals the healthy or coping capacity of the ecosystem to host and quickly recover biodiversity [53,70].

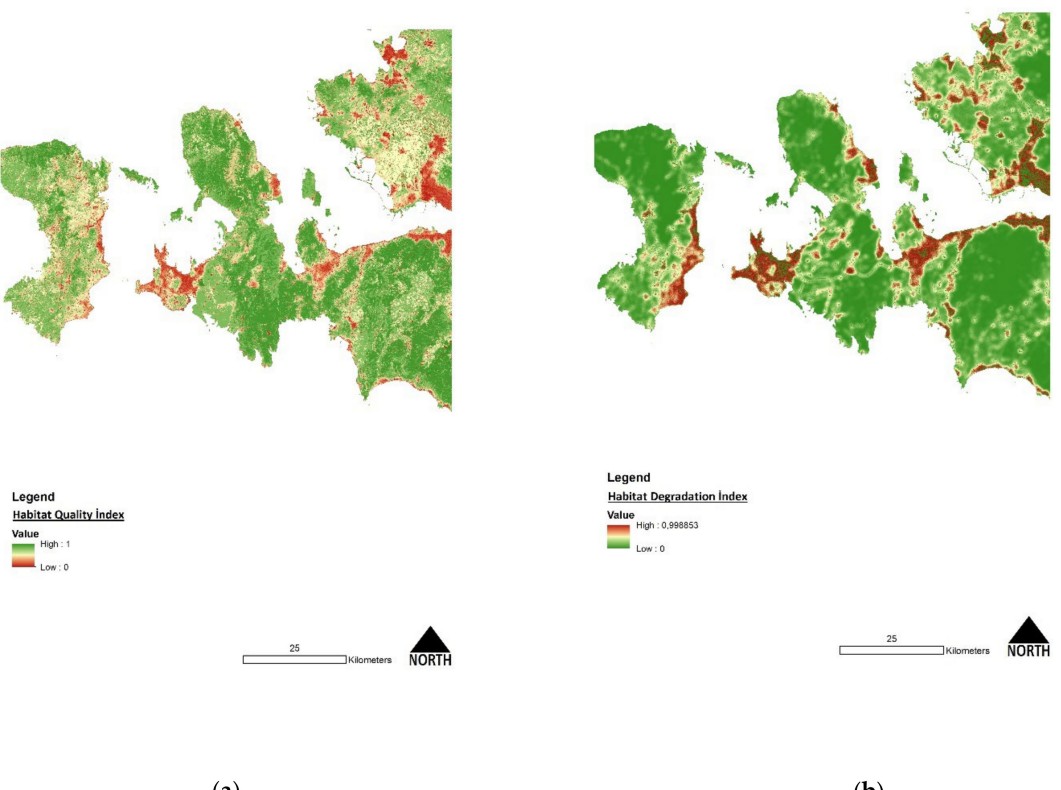

(**a**)                                        (**b**)

**Figure 7.** Habitat Quality output. (**a**) habitat quality index. (**b**) habitat degradation index.

Averagely, in the observed catchment, the Habitat Quality index is 0.67 with a standard deviation of 0.27, while the average Decay is 0.10 with a standard deviation of 0.14. These results, when compared to similar research previously developed in other cities [71–74], demonstrates that the average Habitat Quality is far but, at the same time, even the Decay is higher, too (0.10 instead of 0.03). Nonetheless, the Decay value demonstrates how this environmental mosaic is threatened by anthropic structures that generate a slow-burning pressure on the ecosystem. Even the NDVI seems to be different from previous researchers [53]. Nevertheless, it should be considered that NDVI is a typical seasonal indicator, thus revealing the foliage status, and the two images are taken in different periods (March and September).

The Ecosystem Vulnerability index has been finally composed by an "if–then" concatenation of values among the three biophysical values in the catchment:

- Critical, the mean quality is below the average while the mean decay is above;
- Threatened, the mean quality is above the average and the mean Decay too;
- Low Biodiversity (Low_Bio), the condition is "fair" but NDVI is below the average;
- Fair, the mean quality is above the average while the mean Decay is below;
- Health, the condition is "fair" (not critical nor threatened) and NDVI is above the average.

The value has been represented using the hexagonal grid of ESRI ArcGIS using the "tessellation" tool (see Figure 8).

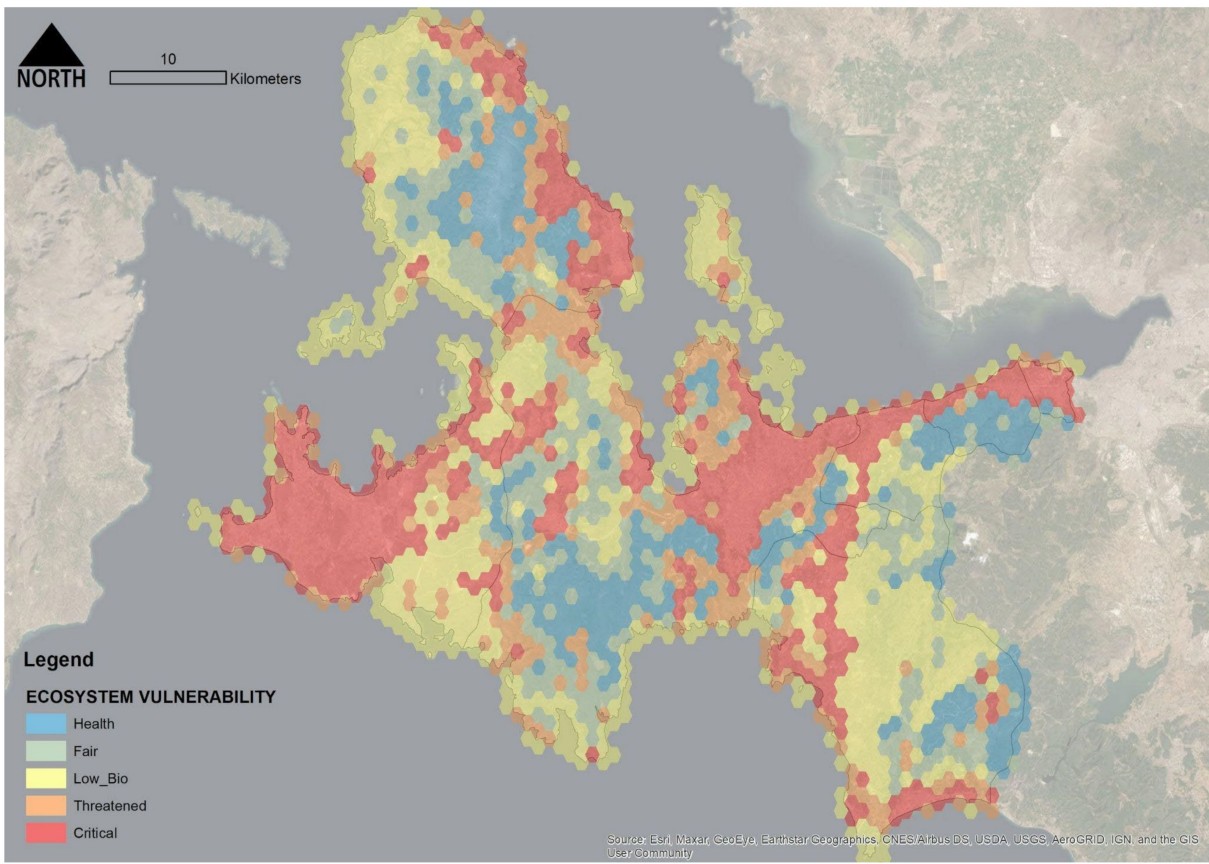

**Figure 8.** Habitat Vulnerability Map.

The map represented in Figure 8 clearly shows how much the recent settlement expansion has shaped the habitat pattern. The west–east coastal conurbation from Izmir, Güzelbahçe, Urla and Seferhisar represents a unique linear cluster of habitat deterioration. The impact of the Cesme touristic area covers entirely the western part of the peninsula, and the eastern coastal part of Karaburun is almost in the same critical situation. On the

borders of the critical patterns, many "transition" zones are threatened by the built-up edge effect, thus receiving the direct disturbance of settlements and infrastructures. On the other hand, it is noteworthy to see how much the Healthy Ecosystems are isolated and disconnected from each other, representing scattered marginal stains of core ecological values that should be adequately protected and valorized through ecological connections.

### 3.2. Conflicts and Synergies with ES Vulnerability

Two layers of the ERP were superimposed to the Ecosystem Vulnerability output while analyzing their distribution and evaluating the results' conflicts and synergies. The layer of the "Conservation Zones" was considered to check if it fits with the real-time biophysical structure of the catchment and, if not, which solutions/modifications/adaptations should be considered (e.g., the extension of the conservation sites or the replacement of the expansion zones). Findings were commented on by using the intersection tool of ESRI ArcGIS while categorizing the output. Therefore, the first layer of Conservation Areas has been classified as follow:

- Health—no conflict but high synergy;
- Fair—no conflict but synergy;
- Low biodiversity—conflict, some action needed to mitigate the effect of urbanization;
- Threatened—moderate conflict action needed (compensation with afforestation action needed);
- Critical—high conflict strong action needed (De-sealing, reducing urban footprint and afforestation).

The first finding suggests that the three categories of conflicts occupy the most significant part of the conservation areas (67% of the conservation areas conflict with the ES vulnerability assessment, see Figure 9). All conservation sites along the Aegean coast contrast with the real ecosystemic condition, thus emphasizing an important environmental degradation process in the protected zones. This peculiar condition demonstrates how the protected areas designed by the ERP fail to address a real tangible barrier against anthropic pressure and ecosystemic decay. The peri-urban areas of Güzelbahçe, Urla, Gülbahçe, the Karaburun promontory, Ildır, Alaçatı and Çeşme are all subject to this conflictual situation: they are included in protected areas to avoid the further expansion of settlements, but they are still involved in processes of densification thus exacerbating a high threat to the ecosystems while being saturated and sealed (see Figure 10) by formal and informal settlements after the 1970s.

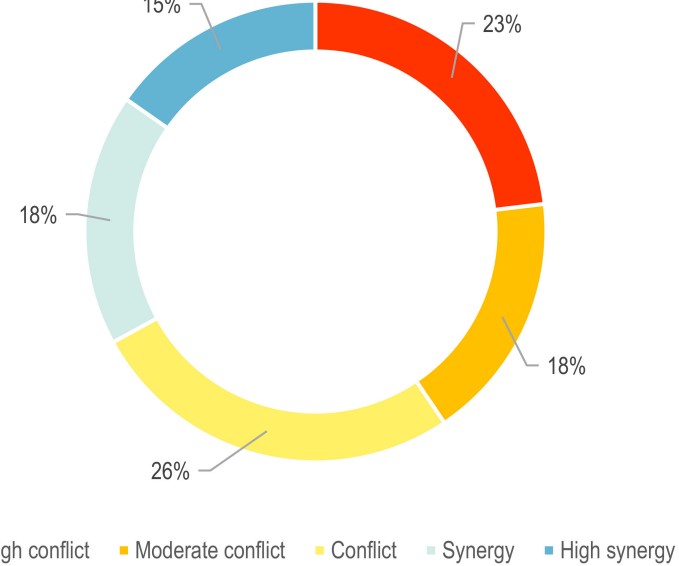

**Figure 9.** Distribution of Conflict and Synergies in the Conservation Zones of the Environmental Regulation Plan.

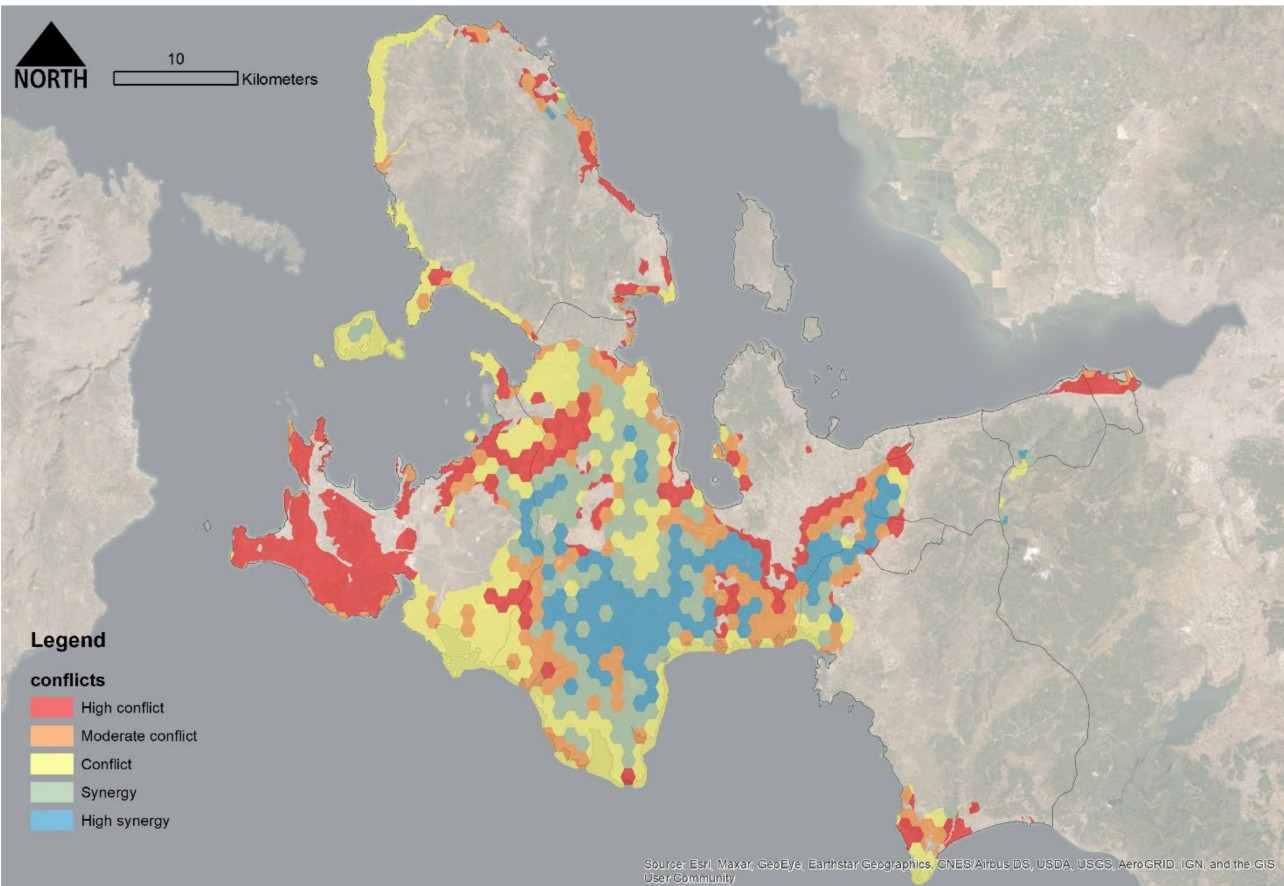

**Figure 10.** Conflicts and Synergies with the Conservation Zones of the Environmental Regulation Plan.

On the other hand, 33% of the conservation areas display synergy with the ES vulnerability; therefore, their inclusion in the protected areas is reasonable and should be even extended to all those areas that are in the same ES condition (health or fair) but are not included in the conservation areas. The idea behind this concept is that all those areas that display a healthy ES status should be strictly regulated and protected against any kind of urbanization process to maintain biodiversity and produce supporting regulative, productive and cultural ES.

The ES Conflict and Synergy analysis can support specific policy orientations toward more sustainable management of the environment while using the categorization as a guideline to address environmental planning actions. In general, strong efforts to control land taken for new urbanization should be taken while applying Urban Growth Boundaries to the Izmir Metropolitan Area [59,74] and introduce even a further land taxation system that considers the environmental degradation of all those urban transformations that happen outside of the urban grow borders identified by the ERP.

Notably, some efforts to introduce an "urban sprawl control" method were introduced by the Izmir Metropolitan Municipality's 90 min intermodal transport incentive (limitless use of public transport in one ticket). The solution may accompany the polycentalization process withcapillary and intermodal public transport accessibility while introducing green growth (including development quality above strict control).

## 4. Discussion

### 4.1. The Need for Updated and Detailed Environmental Spatial Data

As is briefly discussed, the urbanization process in the area has led to the emergence of environmental conflicts due to the location selection of tourism development, energy and mining investments, aquaculture industries and residential development.

Concerning the Karaburun SEPA, which plays a significant role in maintaining the biodiversity of the Izmir–Manisa region, the land-use decisions of the ERP contradict with actually existing development related to wind farms which threaten the protection of this vulnerable ecosystem. The Cesme Tourism Conservation and Development Project is an area regulated by a special law that was announced in 2019. Offering tourism development in a qualified natural projection site, borders of the project area are included in the ERP, whereas land-use decision was not included. In this example, it can be said that this law might become a tool for unregulated development through bypassing plans without considering the environmental characteristics of the area if there are decisive restrictive measures and plan enforcement. The second category covers conservation sites such as natural, historical, archaeological and urban sites. However, the changed status of natural sites by Principal Decision in 2019 was not included in the latest plan revision in 2020. The plan report of the ERP suggests that these areas should be conserved and used following the principal decision. However, this is against the Regulation of Preparation of Spatial Plans (2014).

The conflicts and synergies between the approved ERP and the ecosystem status are discussed to clarify how the ERP should be evaluated and integrated by a real-time ES monitoring assessment (Figure 9).

### 4.2. Ecosystem Service Assessment to Support the Karaburun SEPA Implementation

The presented analysis demonstrates that environmental protection only partially relies on the formal planning system and regulation through an ERP that is applied by a cascade approach. Despite the potential integrations that can better define the new borders and the zoning content of the Conservation Areas, unfortunately, the Turkish planning system is characterized by a high divergence between the planned contents and the real urban dynamics. High negotiation, a poor culture of the formal legitimacy of Plans and Projects and the augmented pressure of the real-estate operators in a growing market are weakening the possibility of pursuing a structured and consistent environmental action. Often, plan amendments fuel unplanned development in vulnerable ecosystems such as the study area and upper-scale spatial decisions bypass environmental regulation plans and protected area borders such as in the Cesme Tourism Conservation and Development Plan.

In this research, the ecosystem service assessment results indicate that, especially in the Karaburun Peninsula, the need to include a new conservation zone in the ERP is a priority. In this area, it is expected that the announcement of it being a SEPA would lead to the protection of Karaburun. However, despite this area being an economically vulnerable cluster, the promontory shows the highest rate of increase in population, at 286% in the last 20 years [63]. During this period, secondary housing development has stretched along the shoreline of Karaburun, where topography allows settlement. In recent work, Cive and Avar (2019) discuss these developments as an example of the commodification of nature under neoliberal urban policies [75]. According to the authors, the common natural resources of local people such as pastures, forests, coastal waters, agricultural lands and natural protected areas have been marketized for the establishment of private investments such as industrial olive production, wind farms, fish farms, quarries and secondary houses. There have been efforts to announce Karaburun as a "Biosphere Reserve Area" among researchers, NGOs and local inhabitants to preserve the biodiversity and habitat integrity of the area to achieve sustainable protection. Yet, currently, there is no status given to the area in this regard [2,76].

For this area, we can suggest here to frame the future environmental regulation considering the SEPA guidelines and use a robust analytical ecosystem assessment to support the identification of the Nature Conservation Zones.

Rather than ambiguous land-use decisions such as "Preferred Choice Land Use", more decisive land-use types are required in ERP regulation. As mentioned, at least some internal sub-zones should be regulated by specific aims and goals: where ES conflict is detected, land-take limitations should be prescribed (e.g., Urban Growth Boundaries); when moderate conflict is detected, some environmental compensation actions should be introduced (e.g., land control for new development, afforestation and greening); where high competition has been detected, the reduction of anthropic pressure should be reduced with land reclamation and challenging de-sealing projects (e.g., de-sealing and NBS).

*4.3. General Implications and Policy Suggestions*

Structural environmental measures should be grounded on a clear, comprehensive spatial strategy for ecosystem protection. According to the site-specific findings of this study, we want to elaborate further on some general policy guidelines that can also be implemented in other contexts, having elaborated the Ecosystem Vulnerability analysis. In particular, ecosystem vulnerability classifications can be widely used in different study contexts to set policy targets and define site-specific regulations. The conservation areas can be regulated through specific regulatory zoning and local planning activity to apply general environmental recommendations:

1.  Where conflict is detected, environmental actions are needed to mitigate the effect of urbanization. The guidelines to limit, mitigate or compensate soil sealing [77] provides a wide extent of examples to see how the impact of land transformation to the ecosystem can be minimized while adopting green mitigative solutions. These solutions rely on the possibility to mitigate (if the transformation is necessary) the direct impact of soil sealing [78] by adopting design and technological solutions that facilitate the permeability of the soil, the shadowing, the presence of green and the concentration of the built volume to reduce impacts on the environment [79–81].

2.  Where moderate conflict is detected, specific green compensation and afforestation actions are needed while re-balancing the ecosystem status affected by soil inefficiencies due to the pressure of the anthropic system [9,40,82]. These actions include typical agro-environmental afforestation policies to protect core ecological areas, minimize urbanization's impact and regenerate biodiversity in degraded places [83–85]. These areas should be selected to prioritize natural landscape intervention to connect ecological corridors and existent biodiversity core sites.

3.  Where high conflict is detected, the ES condition is already alerting and seriously compromised; thus the priority is to reduce the anthropic pressure through a range of complicated ecological interventions. Here the de-sealing, building replacement, the reduction of the urban footprint and the adoption of costly Nature-Based Solutions [86–89] are needed to decrease the impacts of urbanization. The ERP should pay great attention to these areas while amending local masterplans by introducing greening planning solutions to the built environment and demonstrating the tangible benefits that Performance-Based Planning Solutions [90,91] can provide with quantitative assessment.

4.  Lastly, the perimeter of conservation areas should include all those healthy ecosystems that are not protected by any planning document but are pivotal to regenerating the Natural Capital in the catchment (see Figure 11). The new 47,738.71 ha of conserved areas should be included in the revised perimeter of the Conservation Zones of the ERP. These new areas should be only "conserved" while maintaining the existing biodiversity [83,92–94]. Figure 11 clarifies that the Karaburun promontory is largely "uncovered" by special protection, thus requiring immediate intervention, as other already approved documents have already clarified.

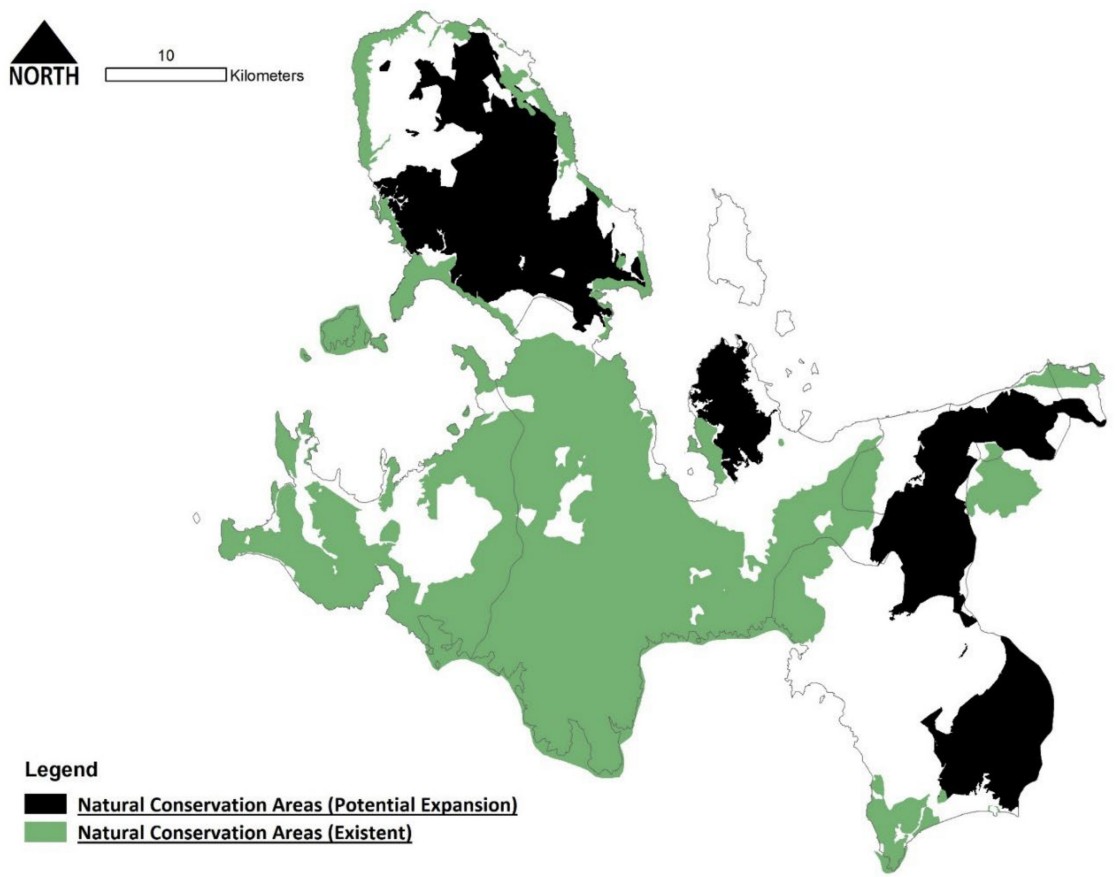

**Figure 11.** The potential inclusion of new Conservation Zones (black) in the Environmental Regulation Plan.

The abovementioned recommendations (green mitigative solutions, compensation and afforestation and re-naturing solutions) can be taken as a replicable example of the kind of actions that are needed to cope with different degrees of ecosystem degradation. Similar conclusions are drawn in other studies where the ecosystem vulnerability assessment has been employed to support the kind and typology of nature-based intervention [95,96]. The study improves and further elaborates a similar index previously developed in Turin (Italy) [54], thus providing some basis for potential replications in other areas threatened by similar settlement expansion processes. Besides, all the inputs of the ecosystem vulnerability map can be accessed and processed by open-access websites (Copernicus Program), ensuring the full replicability of the same ecosystem diagnosis in every part of the world. As for the multiple potential applications, recently, ecosystem vulnerability has been used not only for the conflict and synergy analysis [5] but also for land suitability assessment [97,98], management of primary production [99], sprawl control [83,100], urban growth boundary definition [59,101,102] or land taxation purposes [78]. Therefore, this method can support several generalized implications by its wider application in landscape planning and management at different scales.

## 5. Conclusions

Within this paper, we tried to demonstrate how the detailed spatial analysis of ES by biophysical modelling can effectively support the decision making concerning the Environmental Regional Planning scale by assisting the definition of conservation areas. In doing so, we tried to give an overall picture of how the land-use regulatory framework in Turkey is extremely complicated, not only because it is composed of several hierarchical plans, but also because the coordination, implementation and modification of these plans are subject to large inefficiencies.

The level of the so-called "comprehensive landscape management", pursued by the ERP, which ranges on a scale between 1/100,000 and 1/25,000, is not able to pose an effective and severe limitation for the development of local master plans, at least for what concerns the environmental protection and the real capacity to protect the existent ecosystems [103]. As revealed by our analysis, a considerable amount (67%) of the actual conservation areas displays conflicts with the ES vulnerability assessment. This means that the regional scale of ecological conservation areas does not consider the real biophysical values of the environment. Therefore, these plans have weak probabilities to determine a real and tangible effect on maintaining and conserving the environmental biodiversity of territories.

Besides, as argued in the introduction, the Turkey context is rapidly changing. Fast-growing rates concern both the process of anthropization and the process of population growth. In the Izmir metropolitan city, these processes are accompanied by the re-localization of citizens in the broad metropolitan system due to the new high accessibility provided by the new motorway.

Without a clear vision of a multicentral system, the settlement expansion is occurring everywhere, thus creating an enormous threat to the environment and generating a system largely dependent on commuting—demanding private accessibility.

Unfortunately, the ERP has been subject to numerous revisions and adjustments due to its operational capacity to steer urban development. Indeed, the Plan allows an ex-post legitimacy to unplanned transformations while increasing the deregulatory approach of complex socio-ecological and economic systems.

We also demonstrated how some alternative initiatives are considered in the absence of a proper legal framework to conserve fragile ecosystems (the Karaburun Peninsula) (SEPA). Still, these cannot constitute real alternatives to the regional protection of the environment through the ERP.

The efficacy of urban planning documents should be measured by a higher convergence between the planned contents and the real urban dynamics. As earlier mentioned, this process does not only rely upon the technical capacity of producing plans and projects but on a diffuse culture of legality and a higher capacity to steer the real-estate market by sustainable options through green growth with green building regulations, concentrating the settlement areas and their accessibility while defining a structural environmental network.

**Author Contributions:** Conceptualization, S.S. and S.Ö.-Ş.; data curation, S.S.; formal analysis S.S. and S.Ö.-Ş.; investigation S.S., S.Ö.-Ş. and K.V.; methodology S.S. and S.Ö.-Ş.; supervision K.V.; validation S.S., S.Ö.-Ş. and K.V.; visualization S.S. and S.Ö.-Ş.; roles/writing—original draft, S.S. and S.Ö.-Ş.; Writing—review and editing S.S., S.Ö.-Ş. and K.V. All authors have read and agreed to the published version of the manuscript.

**Funding:** This research received no external funding.

**Institutional Review Board Statement:** Not applicable.

**Informed Consent Statement:** Not applicable.

**Data Availability Statement:** Not applicable.

**Conflicts of Interest:** The authors declare no conflict of interest.

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
