# Peer review of "Integrating Ecosystem Vulnerability in the Environmental Regulation Plan of Izmir (Turkey)—What Are the Limits and Potentialities?"

_urbansci, doi:10.3390/urbansci6010019_

Round 1
Reviewer 1 Report
The article Integrating Ecosystem Vulnerability in the Environmental Regulation Plan of Izmir (Turkey). What limits and potentialities? analyzes the ecosystem vulnerability of the western metropolitan area of Izmir with a focus on spatial planning practices.
In order to publish this article, it can be improved:
Q1. For figure 1, add decimals so that the Other Streets and Paved Roads categories no longer appear by 0%.
Q2. For Figure 6 please change that NDVI_1.tif name to NDWI (these abbreviations used during modeling are relevant to the authors at the time of analysis but must have relevant names in the article).
Q3. The same requirement applies to Figures 7. (quality_c.tif and deg_sum_c.tif).
Q4. For table 3 and the rest of the tables where you used abbreviations (for example Max_Dist), below the tables add explanations for the abbreviations used.
Q4. Line 388 Low_Bio?
Q5. Figure 8 needs some guidance (there may be some important site names, especially those in the Critical vulnerability class.
Q6. The same observation applies to Figure 9. Please remove from the Legend all other values class. It cannot remain active in a map made by GIS experts in a scientific article.
Q7. Figure 11 has the same problem: Lack of label for presented areas and erroneous names in the legend (NC_exp4 and Natural_Conservation_WGS84_2). Please complete and change the names.
Q8. For non-English bibliographic articles and books. Translate and place these bibliographic titles in the format required by the journal.
Best regards.
Author Response
The article Integrating Ecosystem Vulnerability in the Environmental Regulation Plan of Izmir (Turkey). What limits and potentialities? analyzes the ecosystem vulnerability of the western metropolitan area of Izmir with a focus on spatial planning practices.
In order to publish this article, it can be improved:
Q1. For figure 1, add decimals so that the Other Streets and Paved Roads categories no longer appear by 0%.
Thank you for your comment. Now we changed the Figure accordingly.
Q2. For Figure 6 please change that NDVI_1.tif name to NDWI (these abbreviations used during modeling are relevant to the authors at the time of analysis but must have relevant names in the article).
Thank you for the observation, we changed the label. Please just consider that we used a Vegetation Index and not a Water Index; thus the abbreviation is NDVI and not NDWI.
Q3. The same requirement applies to Figures 7. (quality_c.tif and deg_sum_c.tif).
Thank you for your comment. Now we changed the Figure accordingly.
Q4. For table 3 and the rest of the tables where you used abbreviations (for example Max_Dist), below the tables add explanations for the abbreviations used.
Thank you for your comment. Now we changed the Figure accordingly.
Q4. Line 388 Low_Bio?
We apologize for this inconvenience, we forget to remove the abbreviation, it stays for “Low biodiversity”
Q5. Figure 8 needs some guidance (there may be some important site names, especially those in the Critical vulnerability class.
Thank you for your observation, now we added a specific section after the Figure 8, which comments the distribution of the ES vulnerability classes.
Q6. The same observation applies to Figure 9. Please remove from the Legend all other values class. It cannot remain active in a map made by GIS experts in a scientific article.
Thank you for your comment. Now we changed the Figure accordingly.
Q7. Figure 11 has the same problem: Lack of label for presented areas and erroneous names in the legend (NC_exp4 and Natural_Conservation_WGS84_2). Please complete and change the names.
Thank you for your comment. Now we changed the Figure accordingly.
Q8. For non-English bibliographic articles and books. Translate and place these bibliographic titles in the format required by the journal.
The translation of Non-English bibliographic materials are placed in brackets next to titles in the original language.
Best regards.
Reviewer 2 Report
I have enjoyed this opportunity to review this manuscript which addresses the critical issues and opportunities of environmental planning in the case of the city of Izmir.
The manuscript as it is, I believe needs some improvements before being considered for publication. I hope that my suggestions below provide some guidance for the authors as they revise this manuscript.
The abstract with those interrogative sentences is a bit strange. I would recommend using more scientific language.
The manuscript could be of great significance but is quite confusing, it devotes few lines to theoretical background and probably too many to the description of the case study. As I read the introduction I was left wondering what is the literature references of this manuscript? The introduction should provide an overview of the manuscript, including the structure of it. A brief literature review could be also included in the introduction or a specific section can be devoted to it.
In this case the introduction provides some general information about the case study that can be moved in a specific paragraph in material and methods related to the description of the case study.
As for the presentation of the case study I am wondering if all these informations on page 6 are necessary for the purpose of the article. Perhaps they can be simplified or collected in tables.
The results seem interesting but the method does not allow an exhaustive evaluation as it is not explained in a coherent way. In the result section some logical steps are missing.
The results paragraph is really short. It seems to me that paragraphs 4.1 and 4.2 should be moved into the results section. These two paragraphs clearly present the results of the study with data, percentages and maps: why did the authors include these two paragraphs in the discussions?
I would advise the authors to radically review the structure of the article as methods, results and discussions are not consistent with each other.
The discussion and conclusions are very specific to Izmir, and in my mind, could have much better linkages back to the more general literature, which would provide more generalised implications and uses for your methods.
I suggested moderate English language changes, however, the work needing to be done falls between this and minor spell-check. There are some awkward phrasings throughout that should be fixed. I suggest that a native English speaker language proof the manuscript before publication.
Author Response
I have enjoyed this opportunity to review this manuscript which addresses the critical issues and opportunities of environmental planning in the case of the city of Izmir.
Thank you for your comment.
The manuscript as it is, I believe needs some improvements before being considered for publication. I hope that my suggestions below provide some guidance for the authors as they revise this manuscript.
The abstract with those interrogative sentences is a bit strange. I would recommend using more scientific language.
We apologize for that; we probably abused a “personal” style while outlining the research questions directly.
Now we turned the question into basic sentences.
The manuscript could be of great significance but is quite confusing, it devotes few lines to theoretical background and probably too many to the description of the case study.
Thank you for this observation, we tried to re-organize the description of the case of study and work on a broader referenced introduction to the theory on ecosystem and ecological planning (see also point below).
As I read the introduction I was left wondering what is the literature references of this manuscript? The introduction should provide an overview of the manuscript, including the structure of it. A brief literature review could be also included in the introduction or a specific section can be devoted to it.
Thank you for this observation. According to your comment, we decided to revise the chapter “1.3 Aims of this study” as follows (please consider that this is an experimental approach; thus, there aren’t specific manuscripts that deal with the same approach):
- We added much more references to the existing literature of the same field of research;
- We declared that ecosystem mapping assessment in Tukey is not yet currently practised due partly to the paucity of production and distribution of digital data for soil ecosystem mapping and partly due to a weak cultural approach to integrating ecological values in planning;
- We define the novelty of the approach;
- We explicitly refer to the work of Costanza et al. (2017) where the revision of the ES cascade model shows how spatial modelling of the ecosystems constitutes a basic pre-condition for achieving the sustainability of plans and projects, at different scales
In this case the introduction provides some general information about the case study that can be moved in a specific paragraph in material and methods related to the description of the case study.
Thank you for your observation, we moved the introductive part that deals with the land-use change dynamic of the AoI in the methodological part.
As for the presentation of the case study I am wondering if all these informations on page 6 are necessary for the purpose of the article. Perhaps they can be simplified or collected in tables.
Thank you for your comment. Ä°nstead of a table (we are just introducing the context with a discursive style), we reduced the redundant, unnecessary information as suggested.
The results seem interesting but the method does not allow an exhaustive evaluation as it is not explained in a coherent way. In the result section some logical steps are missing.
Thank you for your observation, we tried to re-edit the body of the text more coherently and logically, especially linking more the method with the results, while shortening and focusing the discussion only to the policy suggestions (see point below).
The results paragraph is really short. It seems to me that paragraphs 4.1 and 4.2 should be moved into the results section. These two paragraphs clearly present the results of the study with data, percentages and maps: why did the authors include these two paragraphs in the discussions?
Thank you for this observation. Now we moved some parts of these chapters to the result section.
I would advise the authors to radically review the structure of the article as methods, results and discussions are not consistent with each other.
Thank you for your observation; we reviewed the three parts accordingly.
The discussion and conclusions are very specific to Izmir, and in my mind, could have much better linkages back to the more general literature, which would provide more generalized implications and uses for your methods.
Thank you for your observation, now we added chapter 4.3. entitled ‘Policy Suggestions’ where we open up the results to a broader utilization, having performed in different contexts similar analyses.
I suggested moderate English language changes, however, the work needing to be done falls between this and minor spell-check. There are some awkward phrasings throughout that should be fixed. I suggest that a native English speaker language proof the manuscript before publication.
Thank you for this observation. To increase English correctness, we asked our internal Academic Writing Center to review the manuscript entirely (attached the certificate), and we guess this reviewed version is much more consistent.
Round 2
Reviewer 2 Report
I would like to thank the authors for reviewing the manuscript. The overall quality and structure have been greatly improved.
I still think that the literature review is not sufficient in this article. Usually one of the paragraphs of the introduction are dedicated to a short literature review in order to provide a scientific framework in which the article is placed. The manuscript deals only with the case study without any reference to the wider scientific literature and this could be a weakness.
I believe that some small additional changes could make the manuscript clearer and make the structure more coherent.
For example, in the introduction paragraphs 1.1 and 1.2 could be a single paragraph concerning a general framework7overview of the city of Izmir.
Figure 9 has some layout problems.
Discussions are also focused only on the case study, there is no reference to more general and scientific / literature issues. I have already said that the discussion is very specific to Izmir and it could has much better linkages back to the more general literature which would provide more generalised implications and uses for your methods.
Author Response
I would like to thank the authors for reviewing the manuscript. The overall quality and structure have been greatly improved.
Thank you so much for your observation, we are grateful to receive this appreciation.
I still think that the literature review is not sufficient in this article. Usually one of the paragraphs of the introduction are dedicated to a short literature review in order to provide a scientific framework in which the article is placed. The manuscript deals only with the case study without any reference to the wider scientific literature and this could be a weakness.
Thank you for this observation, now we created a specific chapter of the literature review according to your suggestion.
I believe that some small additional changes could make the manuscript clearer and make the structure more coherent.
For example, in the introduction paragraphs 1.1 and 1.2 could be a single paragraph concerning a general framework7overview of the city of Izmir.
Thank you for your observation, we merged the two paragraphs accordingly.
Figure 9 has some layout problems.
Thank you for your observation, the image was oversized. We reduced the layout accordingly
Discussions are also focused only on the case study, there is no reference to more general and scientific / literature issues. I have already said that the discussion is very specific to Izmir and it could has much better linkages back to the more general literature which would provide more generalised implications and uses for your methods.
Thank you so much for your comment. According to your observation, we changed the title of chapter 4.3. as follow: “General implications and policy suggestions” and we emphasized how the study can be replicated in another context while extending its potential implications largely beyond the western Turkey coastal area.